

# Measuring Chern numbers in Hofstadter strips

**Samuel Mugel[1,2], Alexandre Dauphin[1], Pietro Massignan[1,3], Leticia Tarruell[1],**
**Maciej Lewenstein[1,4], Carlos Lobo[2] and Alessio Celi[1⋆],**

**1** ICFO-Institut de Ciencies Fotoniques, The Barcelona Institute of Science and Technology,
08860 Castelldefels (Barcelona), Spain
**2** Mathematical Sciences, University of Southampton, Highfield, Southampton,
SO17 1BJ, United Kingdom
**3** Departament de Física, Universitat Politècnica de Catalunya,
Campus Nord B4-B5, E-08034 Barcelona, Spain
**4** ICREA, Passeig de Lluís Companys, 23, E-08010 Barcelona, Spain

⋆ alessio.celi@icfo.eu

## Abstract

Topologically non-trivial Hamiltonians with periodic boundary conditions are characterized by strictly quantized invariants. Open questions and fundamental challenges concern their existence, and the possibility of measuring them in systems with open boundary conditions and limited spatial extension. Here, we consider transport in Hofstadter strips, that is, two-dimensional lattices pierced by a uniform magnetic flux which extend over few sites in one of the spatial dimensions. As we show, an atomic wave packet exhibits a transverse displacement under the action of a weak constant force. After one Bloch oscillation, this displacement approaches the quantized Chern number of the periodic system in the limit of vanishing tunneling along the transverse direction. We further demonstrate that this scheme is able to map out the Chern number of ground and excited bands, and we investigate the robustness of the method in presence of both disorder and harmonic trapping. Our results prove that topological invariants can be measured in Hofstadter strips with open boundary conditions and as few as three sites along one direction.


# 1 Introduction

Quantum Hall systems and topological insulators are intriguing materials that are normal insulators in the bulk, but present conducting states at their boundary [1–5]. Of particular interest is the fact that both the number of edge states and the system's conductance are quantized [6]. These measurable effects are the striking consequences of the topological properties of the bulk [7,8] and can be understood in terms of Laughlin's pumping argument [9]: a quantized number of charges is transferred from one edge to the other when, after one period of the pump, the magnetic flux is increased by one quantum. This kind of global behavior, known as topological order, has led to a novel paradigm for phases and phase transitions [10,11]. These materials not only have a fundamental interest, but also interesting technological implications. For instance, the quantization of conductance is routinely exploited for metrology applications. Moreover, the fact that the edge states can obey fractional statistics in presence of interactions [12] makes them ideal candidates for topological quantum computation [13–15].

In recent years, quantum simulators have emerged as a powerful tool in the study of topological phases. They give access to clean and highly controllable experimental systems, where our theoretical understanding of topological physics can be benchmarked. Particularly suitable platforms include ultracold atoms [16,17], photonic devices [18], and mechanical systems [19]. Exploiting them, one dimensional (1D) models [20–28], the Hofstadter model [29–35], and the Haldane model [36] have recently been realized.

Particularly promising systems are *synthetic lattices*, i.e., lattices which have a synthetic dimension, which is obtained by coherently and sequentially coupling particles' internal degrees of freedom. For atoms, this can be done exploiting internal (spin) states, which are coupled using radiofrequency or Raman transitions [37,38]. The latter naturally yield complex tunneling amplitudes, whose phases are linearly dependent on the position of the atoms in real space. This simple construction can be exploited for engineering Hofstadter models out of 1D lattices loaded with spinful atoms [38]. These *Hofstadter strips* present sharp boundaries in the synthetic dimension, making them ideal for the experimental detection

**Sci**|**Post**                                                                    *SciPost Phys. 3, 012 (2017)*

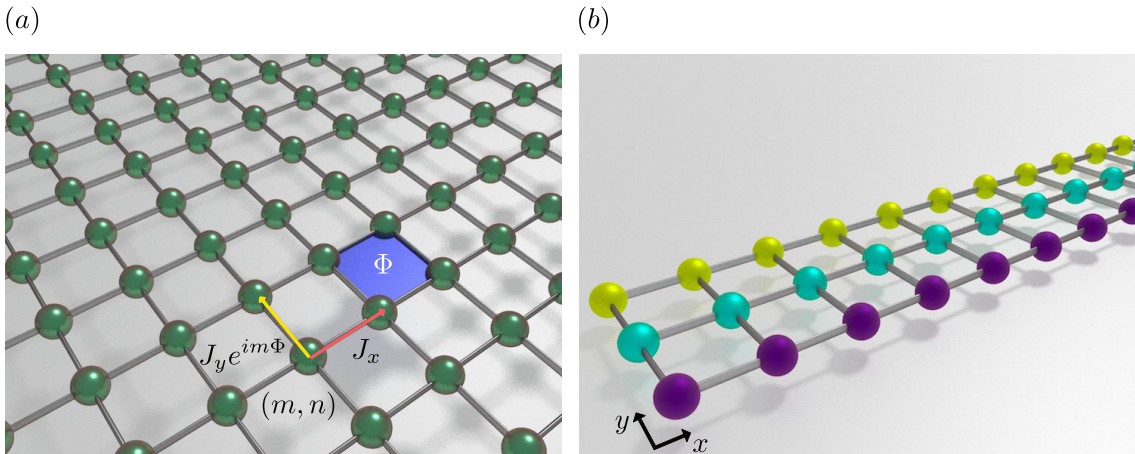

Figure 1: Sketch of the tunnelings of the Hofstadter Hamiltonian, Eq. (1). The site indices in the $x, y$ directions are $m, n$ respectively. The total flux through each plaquette is $\Phi = 2\pi p/q$. Panel (a) depicts the usual 2D Hofstadter extended lattice, while panel (b) shows an Hofstadter strip, which contains few states along the $y$ direction (in the example shown, $N_y = 3$).

of edge states [34, 35, 39]. Synthetic lattices have also been experimentally realized exploiting atomic clock states [40–42], momentum states [25, 43–45], and in integrated photonic platforms [26, 27]. Theoretical proposals have investigated alternative implementations, exploiting modes of harmonically trapped atoms [46] or optical resonators [47–50]. Other studies have considered using them to engineer topological quantum walks [26, 27, 51], study the properties of quantum systems on triangular and hexagonal lattices [52, 53], realize Weyl semimetals [54], simulate 4D models [37] – in particular 4D quantum Hall phenomena [55–57] – or lattices with complex topologies [58]. Finally, these systems open a very promising route to the study of many-body phenomena, especially in topological systems [59–73].

Synthetic lattices are unique in that they present very few sites in one direction. Consequently, one can wonder if the bulk-edge correspondence applies in the Hofstadter strips. It has been shown that edge states may be identified even in systems as thin as a two- or three-leg ladders, where the bulk is either absent, or composed of a single string of atoms [32,34,35,38,74].

In this work we take a complementary point of view, and analyze instead the bulk properties of these Hofstadter strips. We show that they can be probed efficiently by performing Bloch oscillations under the action of a constant force, thereby extending the concept of Laughlin pumping of filled bands to single particle dynamics. In particular, we prove that the transverse displacement of a suitably initialized particle in the Hofstadter strip yields an accurate measurement of the topological properties of the whole band structure. Additionally, we show that the measurement is robust against static disorder and may be reliably performed even inside a harmonic trap.

## 2 The model

### 2.1 Hofstadter model

The Hofstadter model [75], sketched in Fig. 1a, describes non-interacting spinless atoms in a two-dimensional square lattice in presence of a uniform external artificial magnetic field. For an effective flux $\Phi = 2\pi p/q$ per plaquette (with $p$ and $q$ coprime integers), in the Landau

gauge (with the gauge field along $y$), the Hofstadter Hamiltonian on a $N_x \times N_y$-torus takes the form

$$\hat{H}_0 = -\sum_{m,n} J_x \hat{c}^\dagger_{m+1,n} \hat{c}_{m,n} + J_y e^{im\Phi} \hat{c}^\dagger_{m,n+1} \hat{c}_{m,n} + \text{H.c.,} \tag{1}$$

where $\hat{c}^\dagger_{m,n}$ creates a particle at site $(m,n)$, with $m$ and $n$ the site indices in the $x$ and $y$ directions, and $J_x, J_y > 0$ the tunneling amplitudes. Within this gauge the magnetic unit cell spans $q$ sites in the $x$ direction, so that the Brillouin zone has an extension of $(2\pi/qd) \times (2\pi/d)$ along $k_x$ and $k_y$, respectively, where $d$ denotes the lattice spacing. The spectrum of $\hat{H}_0$ presents $q$ bands formed by Bloch eigenstates $|u_j(\mathbf{k})\rangle$ with energies $E_j(\mathbf{k})$ ($j = 1, \ldots, q$). The Hofstadter Hamiltonian breaks time-reversal, particle-hole, and chiral symmetries, and therefore belongs to the unitary class A [76]. In two spatial dimensions, the topology of each energy band $j$ is characterized by a Chern number $\mathscr{C}_j \in \mathbb{Z}$

$$\mathscr{C}_j = \frac{1}{2\pi} \int_{BZ} \mathscr{F}_j(\mathbf{k}) d^2\mathbf{k}, \tag{2}$$

defined as the integral of the Berry curvature $\mathscr{F}_j(\mathbf{k}) = 2\,\text{Im}\langle \partial_{k_y} u_j(\mathbf{k}) | \partial_{k_x} u_j(\mathbf{k}) \rangle$ over the Brillouin zone.

## 2.2 Hofstadter strips

As theoretically proposed in Ref. [38] and recently realized experimentally with ultracold atoms [34, 35, 40–42], the Hofstadter model can be simulated with the help of synthetic lattices. By interpreting the atoms' spin as an extra dimension [37], the experiments realize an elongated strip, subject to a uniform effective magnetic field. We review here the properties of such systems. Let us consider a narrow strip with few sites in the $y$ direction. For an easier analytic understanding, it is useful to perform the gauge transformation $\hat{c}_{m,n} \to e^{imn\Phi} \hat{c}_{m,n}$, which transfers the phases along the $x$-direction, so that the magnetic unit cell extends along the $y$ direction.

The Hamiltonian of Eq. (1) with periodic boundary conditions along $x$ and open along $y$ reads

$$\hat{H}_0(k_x) = -\sum_n 2J_x \cos(k_x d - n\Phi) \hat{c}^\dagger_{k_x,n} \hat{c}_{k_x,n} + (J_y \hat{c}^\dagger_{k_x,n+1} \hat{c}_{k_x,n} + \text{H.c.}), \tag{3}$$

with $k_x \in [-\pi/d, \pi/d]$ the quasi-momentum in the $x$ direction. The dispersion relation of this Hamiltonian is shown in Fig. 2 for $\Phi = 2\pi/3$. For each value of $k_x$, the spectrum is composed of $N_y$ discrete energy values. When the system has periodic boundary conditions along $y$ (dashed gray lines), these are grouped in $q$ distinct bands, separated by band gaps $\Delta E_j$, $j \in [1, q-1]$, and the Brillouin zone has an extension of $(2\pi/d) \times (2\pi/qd)$ along $k_x$ and $k_y$. The solid lines instead depict the spectrum of the system with open boundary along $y$. In Fig. 2a we set $J_y = J_x/5$, and we see the appearance of states which bridge the band gap. Because these are sharply localized at the edges of the system (as indicated by the line coloring), we will refer to them as edge states. In Fig. 2b we choose instead isotropic tunnelings, $J_y = J_x$. Under these conditions the two edge states largely overlap and hybridize, leading to hybridization gaps $\Delta\varepsilon_j < \Delta E_j$.

## 3 Measurement of the Chern number

In ultracold atomic systems, direct measurements of the Hall conductivity are challenging since conventional transport experiments require specific setups [77]. Several schemes have

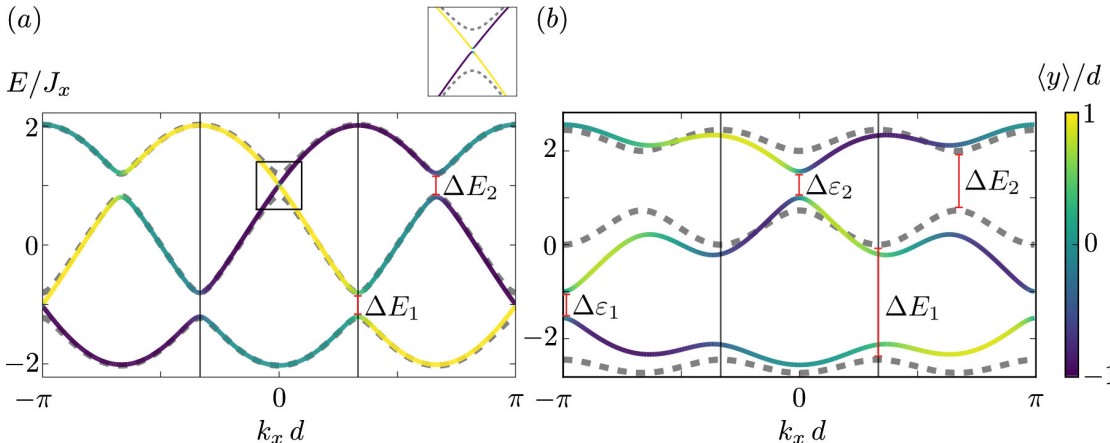

Figure 2: Dispersion of an Hofstadter strip with $N_y = 3$ and flux $\Phi = \frac{2\pi}{3}$, for (a) $J_y = J_x/5$, and (b) $J_y = J_x$. Dashed lines denote the dispersion relation with periodic boundary conditions. Solid lines refer to open boundary conditions along $y$, and their color coding indicates the mean eigenstates' position in that direction. Well localized edge states exist in the gaps of the periodic system, see inset of panel (a). Edge states overlap, giving rise to hybridization gaps $\Delta\epsilon_j < \Delta E_j$, where $\Delta E_j$ denote the band gaps. However, the effect is negligible when $J_y \ll J_x$.

however been developed to determine the Chern number or the Berry curvature of topological Bloch bands, based either on time-of-flight measurements [78–81], or on the dynamics of the center of mass of the cloud [36, 82–85]. In the strip geometry, earlier works proposed to measure the Chern number by means of hybrid time-of-flight and in-situ measurements [86], or exploiting the interplay of magnetic flux and strong interactions [61, 62, 70]. Another way to observe the effects of Berry curvature is to consider 1D systems and to use time as an extra dimension. For example, recent experiments with ultracold atoms realized Thouless pumping [22, 23], spin pumping [87], and geometrical pumping [88].

Here, we propose a simple scheme to measure the Chern number of a Hofstadter strip which requires three main ingredients: the adiabatic preparation of an atomic wave packet in the ground state of the lattice, the application of a force to realize the pumping, and the precise measurement of the center-of-mass in the direction perpendicular to the force. In the following, we specifically discuss the importance of the preparation of the initial state and explain how the force allows one to scan the whole Brillouin zone and to reveal the Chern number. Finally, we exploit the unique feature of synthetic dimensions, namely that the center-of-mass dynamics in the transverse direction can be read out from spin populations. We start by showing that, in a periodic 2D system which presents a filled band, the semi-classical dynamics are determined by the band's Chern number. We then proceed to show that the Chern number can be deduced from the mean displacement even when the band is occupied by an atomic wave packet, for instance a weakly-interacting Bose-Einstein condensate (BEC). We conclude the Section by discussing in detail the experimental procedure needed to implement the proposed method in a Hofstadter strip.

## 3.1  Semiclassical equations of motion and Chern number

Let us consider the Hofstadter Hamiltonian $\hat{H}_0$ with periodic boundary conditions, given by Eq. (1), subject to an external constant force applied in the $x$ direction

$$\hat{H} = \hat{H}_0 - F_x \hat{X} = \hat{H}_0 - F_x \sum_m m\, d\, \hat{c}_{m,n}^\dagger \hat{c}_{m,n}. \tag{4}$$

Translational invariance is broken by the force, which plays the role of an electric field. It can be restored by performing the gauge transformation $\hat{U} = \exp(-iF_x\hat{X}t/\hbar)$. This is equivalent to a time-dependent change of the magnetic flux, which is the situation normally considered in the solid-state context [9]. For a wave packet initially occupying a single band, and in the adiabatic approximation (i.e., no Landau-Zener transitions), the evolution of the momentum of a Bloch state with initial momentum component $\mathbf{k}_0$ is given at the semi-classical level by

$$\mathbf{k}(t) = \mathbf{k}_0 + \frac{F_x t}{\hbar}\mathbf{e}_x, \tag{5}$$

where $\mathbf{e}_j$ is the unit vector, with $j = x, y$. Within linear response theory, the mean velocity of the Bloch state can be written as [8]

$$\mathbf{v}_j(\mathbf{k}) = \frac{1}{\hbar}\partial_{\mathbf{k}}E_j(\mathbf{k}) + \frac{F_x}{\hbar}\mathscr{F}_j(\mathbf{k})\mathbf{e}_y. \tag{6}$$

The first term in Eq. (6) is the band velocity, while the second term is the anomalous velocity, which is proportional to both the force $F_x$ and the Berry curvature $\mathscr{F}_j$. For a filled energy band $j$, i.e., when all its Bloch states are uniformly occupied, the contribution of the band velocity averages to zero, and the velocity of the center of mass becomes proportional to the Chern number in agreement with the celebrated TKNN formula [7],

$$\langle\mathbf{v}_j\rangle = \frac{1}{A_{BZ}}\int_{BZ}\mathbf{v}_j(\mathbf{k})\mathrm{d}^2\mathbf{k} = \frac{2\pi F_x}{\hbar A_{BZ}}\mathscr{C}_j\mathbf{e}_y. \tag{7}$$

Here $A_{BZ} = (2\pi)^2/(qd^2)$ is the area of the Brillouin zone. Inspired by the above discussion, we will now present a method to measure the Chern number from the displacement of an atomic wave packet $|\psi(\mathbf{r}, t)\rangle$. At all times, the velocity of its center of mass is given by

$$\langle\mathbf{v}(t)\rangle = \sum_j\int_{BZ}\mathbf{v}_j(\mathbf{k})\rho_j(\mathbf{k}, t)\mathrm{d}^2\mathbf{k}, \tag{8}$$

where $\rho_j(\mathbf{k}, t) = |\langle u_j(\mathbf{k})|\psi(\mathbf{k}, t)\rangle|^2$ is the probability density that the particle at time $t$ occupies a Bloch state of quasi-momentum $\mathbf{k}$ in the $j^{\text{th}}$ band.

From now on, we consider the special case of a wave packet which is initially strongly localized along $y$, while it presents a rather smooth and extended profile along $x$. In momentum space, the corresponding wavefunction $|\psi(\mathbf{k}, t=0)\rangle$ will be sharply peaked in the $x$ direction around $k_x = 0$, while it will occupy uniformly the Brillouin zone in the $y$ direction. For $t > 0$ we may then write $\rho_j(\mathbf{k}, t) \approx \rho_j(k_x(t))$, which is peaked around $k_x = F_x t/\hbar$. We assume that the wave packet occupies only the $j^{\text{th}}$ band, such that the sum over $j$ disappears.

As long as the characteristic energy associated to the force, $|F_x|d$, is much smaller than the band gaps, interband transitions are strongly suppressed and the quasi-momentum increases smoothly according to Eq. (5). The mean displacement after one period is

$$\langle\Delta\mathbf{r}\rangle \equiv \langle\mathbf{r}(T) - \mathbf{r}(0)\rangle = \int_0^T\langle\mathbf{v}(t)\rangle\mathrm{d}t = \int_{BZ}\mathbf{v}_j(\mathbf{k})\left(\int_0^T\rho_j(k_x(t))\mathrm{d}t\right)\mathrm{d}^2\mathbf{k}. \tag{9}$$

Since the probability density displaces with uniform velocity, the mean Bloch state occupation over a period of the force is simply the uniform distribution,

$$\frac{1}{T}\int_0^T\rho_j(k_x(t))\mathrm{d}t = \frac{1}{A_{BZ}}. \tag{10}$$

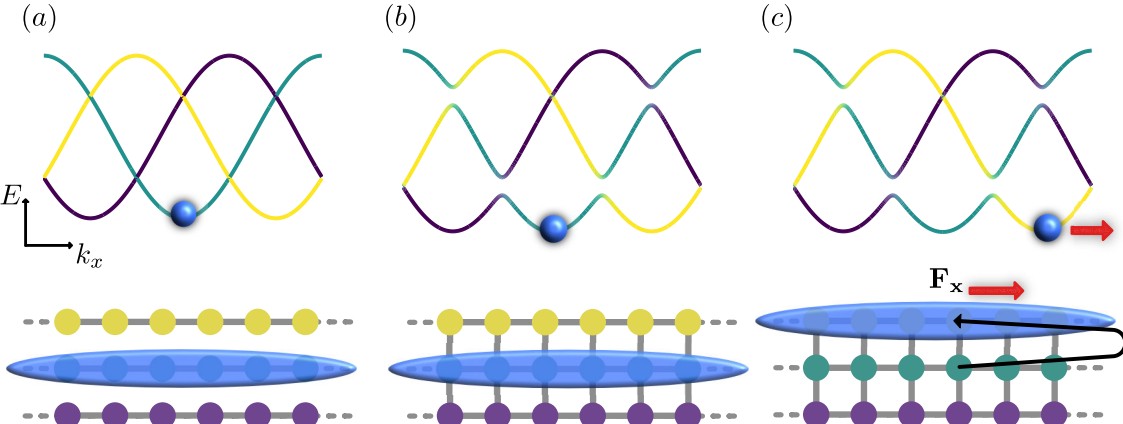

Figure 3: (a) Initially, the Hamiltonian has no tunneling in the $y$ direction. The state is localized at $y = 0$. (b) The tunneling in the $y$ direction is adiabatically turned on, opening gaps in the band structure. (c) Once the adiabatic loading is completed, a force $\mathbf{F_x}$ is applied along $x$, and the wave packet performs Bloch oscillations. The black arrow depicts schematically the motion of the center of mass over the first period of the force. After a complete period, the wave packet returns to the same position along $x$, but it has moved along $y$ of a number of sites proportional to the Chern number of the corresponding energy band.

Substituting in Eq. (9) the general relation Eq. (6), and noting that the contribution from the band velocity cancels over a complete period, one obtains

$$\langle \Delta \mathbf{r} \rangle = \frac{T}{A_{BZ}} \int_{BZ} \mathbf{v}_j(\mathbf{k}) \mathrm{d}^2 \mathbf{k} = \mathrm{sgn}(F_x)\, d\, \mathscr{C}_j\, \mathbf{e}_y. \tag{11}$$

Thus, a constant force along $x$ displaces (*pumps*) particles in the $y$ direction. For the initial state considered here, the number of sites that the atomic center of mass is pumped after a time $T$ is exactly the Chern number [7, 89].

We have derived the mean displacement after one pumping period, given by Eq. (11), in the gauge of Eq. (1), the Landau gauge with the gauge field along the $y$ direction. Naively, we could expect this relation to change if we set the gauge field along $x$ instead. As discussed in Refs. [90–92], due to the symmetry of the lattice, both the energy spectrum and the Berry curvature are completely defined in the so-called reduced magnetic Brillouin zone $-\pi/(qd) \leq k_x, k_y \leq \pi/(qd)$. Hence, during one complete period of the force, $T \equiv 2\pi\hbar/(qd|F_x|)$, the wave packet explores the reduced magnetic Brillouin zone in its entirety, thereby performing a complete Bloch oscillation. This is true regardless of the gauge choice. Thus, we see that the relation between the mean displacement and the Chern number, as given by Eq. (11), is independent of the gauge choice, as is always the case for physical quantities.

## 3.2 Pumping on the Hofstadter strip

In this section, we will explain how the previous discussion can be applied to the Hofstadter strip, which has open boundary conditions and few sites in the $y$ direction. The state preparation is a crucial point in the measurement of the Chern number. The experimental protocol we suggest is summarized in Fig. 3 for the case of a three-leg Hofstadter strip, subject to an external magnetic flux of $\Phi$ per plaquette.

In order for the theory explained above to be applicable, we must ensure that all the dynamics takes place in a single band of the bulk. To this end, a wave packet with the following

properties should be prepared: i) it occupies a small portion of a single band, ii) it is extended along $x$, and iii) it is localized along $y$ (i.e., the wave packet is spin-polarized). The state is prepared in the lowest eigenstate located at $y = 0$ of the Hamiltonian with no tunneling in the $y$ direction (see Fig. 3a).

We then turn on the tunneling in the $y$ direction, linearly from 0 to $J_y$, over a time which is large compared to the inverse of the energy difference $E_{2,1}(k_x = 0) \equiv E_2(k_x = 0) - E_1(k_x = 0)$, such that the process is adiabatic, and the transfer to the other bands is minimized (see Fig. 3b). Since the atomic wave packet always occupies a minimum of the dispersion relation, the center of mass is not displaced during the loading procedure.

At the end of the loading sequence (i.e., once the tunneling along $y$ has reached the desired value), a constant force is applied along $x$, and the resulting dynamics is studied. After a complete period of the force, the quasi-momentum of the wave packet occupies the neighboring minimum of the dispersion relation. In real space this corresponds to a Bloch oscillation along $x$, and a net transverse displacement along $y$ by a number of sites corresponding to the Chern number of the band (see Fig. 3c). Note that if the physical lattice has open boundary conditions along $x$ as well, Eq. (5) is only valid far away from these edges. As such, the lattice should be sufficiently extended along $x$ to accommodate a complete Bloch oscillation.

The different steps of this loading sequence can be readily realized in state-of-the-art experiments. These are very similar to the schemes already used in experiments studying the edge state dynamics of Hofstadter strips [34, 35]. Our protocol requires as additional ingredient a constant force, which should be identical for all the spin states. It can be implemented either by employing a moving optical lattice [93, 94], a linear potential realized optically, or simply the projection of gravity along the lattice direction.

## 4 Pumping dynamics

### 4.1 Chern number measurement with $J_y \ll J_x$

We first consider the three-leg Hofstadter strip ($N_y = 3$) subject to a flux $\Phi = 2\pi/3$ and with a small tunneling ratio $J_y/J_x = 1/5$. The energy spectrum is shown in Fig. 2a and has an energy gap $\Delta E_1 = 0.42 J_x$. Due to the small tunneling ratio, the edge states are each well localized at $n = \pm 1$ and cross at $k_x = 0, \pm \pi/d$. Initially, the wave packet is in the ground state of a box potential with hard walls and extension $w_x = 30$ sites in the $x$ direction, and is fully polarized with $y = 0$. At $t = 0$, a force $F_x = 0.03 \Delta E_1/d$ is applied to the state in the $x$ direction. In this case, the hybridization gap is so small relative to all other energy scales in the system that its effects can be neglected.

Figure 4 displays the results of the simulations: Fig. 4a shows the dynamics of the center of mass, and Fig. 4b the populations of the energy bands/gaps. These were obtained by projecting the evolved state on the eigenvectors of the Hamiltonian, Eq. (3). We define the lower (upper) gap as the separation between bands at $k_x = \pi/3d$ ($k_x = 2\pi/3d$), and assume that whatever lies outside the gaps belongs to the corresponding band.

After one period of the force, the state is pumped from $\langle y \rangle = 0$ to $\langle y \rangle = 0.99$ (from $A$ to $B$), yielding a measured Chern number $\mathscr{C}_1 = 0.99$. This measurement of $\mathscr{C}_1$ agrees extremely well with the value of 1 computed for a system with periodic boundary conditions using the Fukui-Hatsugai-Suzuki (FHS) method [95].

For systems with $J_y \ll J_x$, the Chern number can actually be directly read off from Fig. 2a [7, 96]. Indeed, Eq. (5) shows how the mean momentum is displaced over time due to the force. For any given $k_x$, the color coding of Fig. 2a depicts the eigenstates' mean position along $y$. By combining these two pieces of information, it is therefore possible to estimate the

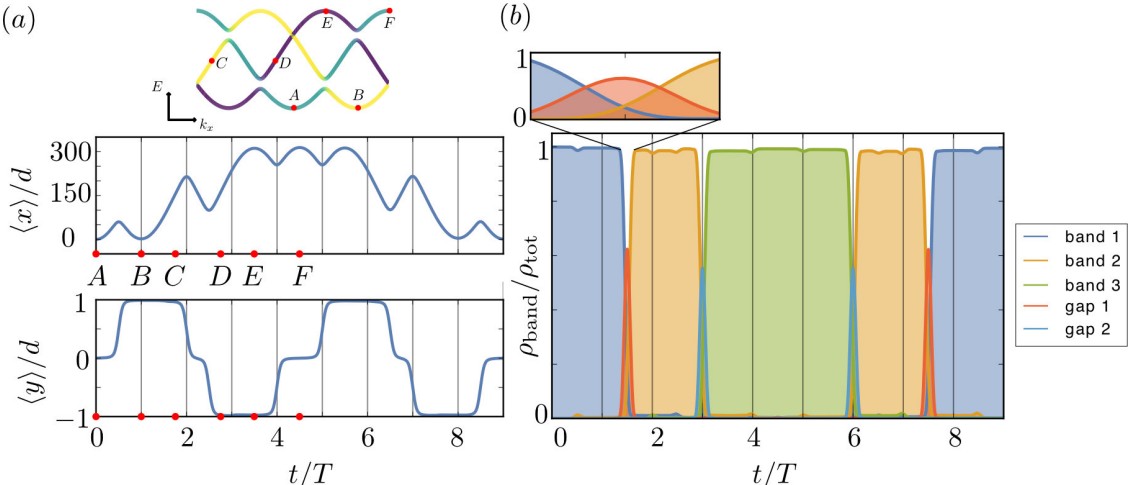

Figure 4: (a) Mean position of the atomic cloud along the $x$ and $y$ directions and (b) populations $\rho$ of bands and band gaps as a function of time, for $J_y = J_x/5$ and $\Phi = \frac{2\pi}{3}$. The red dots labeled $A-F$ indicate the corresponding positions in the dispersion relation, as displayed in the inset of (a). The inset of (b) shows a close-up of the populations in the vicinity of the band crossing. Time is measured in units of the period of the force, $T = 2\pi\hbar/(qd|F_x|)$.

wave packet's mean $y$ displacement over a complete period of the force and extract the Chern number. Related methods to determine the Chern number have been proposed in Refs. [61, 70, 86], where the required quasi-momentum change of $2\pi/d$ was obtained exploiting time-of-flight expansion, a temporal variation of the magnetic flux or a force for a partially filled band.

Once the state reaches the edge along $y$, it is pumped along the edge states to the second band. During this period, the displacement saturates at the edge value $\langle y \rangle \approx 1$. Thus, this measurement of the Chern number is robust to small errors in the determination of the period of the force, or in the preparation of the wave packet. While the edge state is populated, we observe a rapid displacement of the wave packet's center of mass in the $x$ direction. Assuming that there is vanishing overlap between the edge states, we can calculate the mean edge velocity from Eq. (3)

$$\langle v_x \rangle = \frac{1}{\hbar} \frac{\partial E(k_x)}{\partial k_x}\bigg|_{k_x = \frac{\pi}{d}, n=1} \approx \frac{\sqrt{3} J_x d}{\hbar}, \tag{12}$$

which is within 4% of the slope of $\langle x \rangle$ between $t = 3T/2$ and $t = 2T$ (measured from Fig. 4a). At $t = 1.75T$, the mean quasi-momentum of the state is centered at point $C$ in Fig. 2a; it is subsequently pumped to point $D$, which is reached at $t = 2.75T$. We can deduce the Chern number of the second band by measuring the center of mass positions at these times; this provides the estimate $\mathscr{C}_2 \approx -1.94$, which is to be compared with the value of $-2$ given by the FHS algorithm. Subsequently, almost all the density is promoted to the third band along the $n = -1$ edge state. Between times $t = 3.5T$ and $t = 4.5T$, the atomic density is pumped

| Band | Formula | Value |
|------|---------|-------|
| $j = 1$ | $\langle y \rangle_{t=T} - \langle y \rangle_{t=0}$ | $\mathscr{C}_1 \approx 0.99$ |
| $j = 2$ | $\langle y \rangle_{t=2.75T} - \langle y \rangle_{t=1.75T}$ | $\mathscr{C}_2 \approx -1.94$ |
| $j = 3$ | $\langle y \rangle_{t=4.5T} - \langle y \rangle_{t=3.5T}$ | $\mathscr{C}_3 \approx 0.97$ |

Table 1: Chern numbers extracted from the data in Fig. 4.

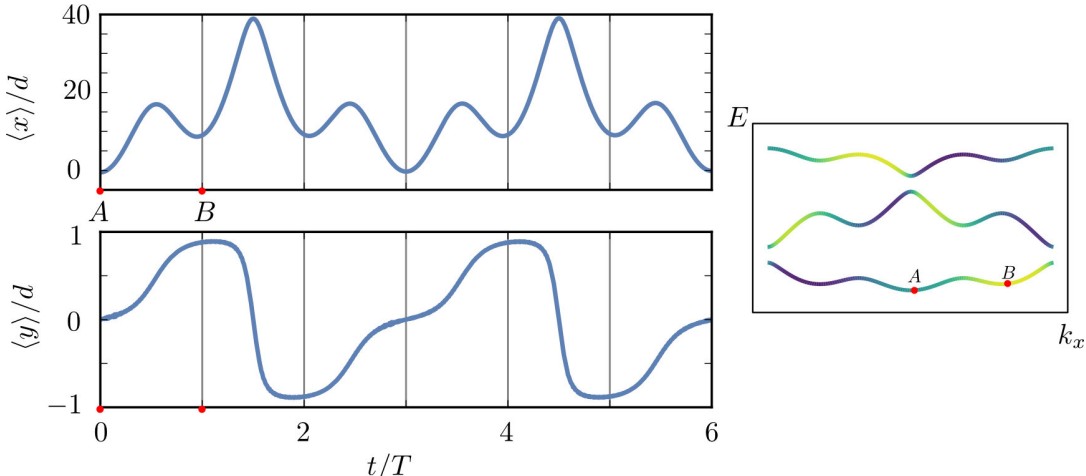

Figure 5: Center of mass position in the $x$ (top) and $y$ (bottom) directions for $J_y = J_x$ and $\Phi = \frac{2\pi}{3}$. During the first period of the force (from $A$ to $B$), the wave packet displaces by approximately one lattice site, in correspondence with the expected value of $\mathscr{C}_1 = 1$. Due to the large hybridization of the edge states, the dynamics is always constrained to the ground band, and it is therefore periodic with period $3T$.

from point $E$ to $F$ in Fig. 4a. As previously, we estimate the Chern number of the third band, yielding: $\mathscr{C}_3 \approx 0.97$, in good agreement with the value 1 given by the FHS algorithm. These measurements are summarized in Table 1. A total of $qN_y = 9$ periods are necessary for the system to return to its initial state.

## 4.2 Chern number measurement with $J_y = J_x$

We now consider a three-leg Hofstadter strip ($N_y = 3$ sites in the synthetic direction) with a tunneling ratio of $J_y/J_x = 1$. The dispersion relation for this system is represented in Fig. 2b; in the presence of periodic boundary conditions, the lowest energy gap equals $\Delta E_1 = 2.45J_x$. When the system has open boundary condition and isotropic tunneling amplitudes, the edge states are not strongly localized at the edge of the system and are able to hybridize, yielding a strong modification of the band structure (compare colored lines with gray ones in Fig. 2b). Initially, the state is confined to a region $w_x = 30$ sites wide, and is fully polarized (i.e., it occupies only the $y = 0$ sites). Then, the tunneling along $y$ is turned on linearly, and a force $F_x = 0.01\Delta E_1/d$ is applied to the state in the $x$ direction. Note that, due to the increased first band gap, this force is much larger than the one considered in Sec. 4.1.

The dynamics of the center of mass is presented in Fig. 5. In the first period of the motion (i.e., from $A$ to $B$), the center of mass is pumped to the $n = 1$ edge, resulting in the mean displacement $\langle y(T) \rangle - \langle y(0) \rangle = 0.88$. Thus, the measurement, although still quite accurate, underestimates the Chern number. In the next subsection, we will show that this deviation is mainly caused by the delocalization of the edge states in spin space.

## 4.3 Effect of the tunneling ratio on the Chern number measurement

As we have seen in the previous subsection, the accuracy of the method decreases with increasing $J_y/J_x$. We now discuss the effects which affect our measurement of the Chern number as we move away from the $J_y \ll J_x$ limit. We identify the hybridization of spin states, i.e., the fact that the Hamiltonian's eigenstates become delocalized in spin space, as the main cause of

deviations of the measurement from the ideal result, at least when the assumption of adiabatic pumping is valid. In the following, we first calculate the mean atomic displacement over one period using perturbation theory, which we then compare to numerical results.

### 4.3.1 Bloch wavefunctions to second order in perturbation theory

By treating $\lambda \equiv J_y/J_x$ as a small parameter, we can use perturbation theory to find the approximate eigenstates of $\hat{H}_0(k_x)$, given by Eq. (3). For simplicity, we restrict ourselves to the case $N_y = 3$ and $\Phi = 2\pi/3$. When $J_y = 0$, the eigenstates of $\hat{H}_0(k_x)$ are simply given by $|n\rangle$ for all $k_x$, i.e, the spin polarized state, with $n \in \{-1, 0, 1\}$. For arbitrary $J_y$, let $|\varphi_n(k_x)\rangle$ be the eigenstate of $\hat{H}_0(k_x)$ which converges to $|n\rangle$ in the limit $J_y \to 0$.

We initiate the system in the state with well defined momentum $|\Psi_0\rangle = |\varphi_0(k_x = 0)\rangle$, which is an eigenstate of $\hat{H}_0$ whose eigenvalue corresponds to point $A$ in Fig. 4a. After a period of the force, the atomic wave packet is adiabatically pumped to $|\Psi_T\rangle = \left|\varphi_1(k_x = \frac{2\pi}{3d})\right\rangle$ (point $B$). The total displacement in the $y$ direction over one period of the force is therefore simply given by

$$\langle \Delta y \rangle = \langle \Psi_T | \hat{n} | \Psi_T \rangle - \langle \Psi_0 | \hat{n} | \Psi_0 \rangle, \tag{13}$$

where $\hat{n}$ is the position operator in the spin direction. Note that, due to the symmetry of the Hofstadter strip around $n = 0$, the mean spin of $|\varphi_0\rangle$ is always zero, such that $\langle \Psi_0 | \hat{n} | \Psi_0 \rangle = 0$.

In this setting, it is therefore sufficient to find the approximate expression of the eigenstate $|\Psi_T\rangle$ to find the mean transverse displacement over a period. Because this state does not occur at a degeneracy point, we can do this using non-degenerate perturbation theory. By defining

$$\mathscr{E}_{2,1} = \frac{E_{2,1}}{J_x} = 2\left(\cos(k_x d - \frac{2\pi}{3}) - \cos(k_x d)\right)\bigg|_{k_x = \frac{2\pi}{3d}} = 2\left(1 - \cos\frac{2\pi}{3}\right) = 3, \tag{14}$$

we have

$$|\Psi_T\rangle = \left[1 - \frac{\lambda^2}{2}\left(\frac{1}{\mathscr{E}_{2,1}^2} + 2\mathrm{Re}\langle v|n = 1\rangle\right)\right]\left(|n = 1\rangle + \frac{\lambda}{\mathscr{E}_{2,1}}|n = 0\rangle + \lambda^2|v\rangle\right) + \mathscr{O}\left(\lambda^3\right), \tag{15}$$

where the first factor comes from the normalization and $|v\rangle$ is the (unnormalized) second order correction to $|n = 1\rangle$. Substituting into Eq. (13) and expanding to second order in $\lambda$, we find

$$\begin{aligned}
\langle \Delta y \rangle &= \langle \Psi_T | \hat{n} | \Psi_T \rangle \\
&= \left[1 - \lambda^2\left(\frac{1}{\mathscr{E}_{2,1}^2} + 2\mathrm{Re}\langle v|n = 1\rangle\right)\right]\left(1 + 2\lambda^2\mathrm{Re}\langle v|n = 1\rangle\right) + \mathscr{O}\left(\lambda^3\right) \\
&= 1 - \lambda^2\frac{1}{\mathscr{E}_{2,1}^2} + \mathscr{O}\left(\lambda^3\right)\bigg|_{k_x = \frac{2\pi}{3d}} \\
&= 1 - \left(\frac{J_y}{3J_x}\right)^2 + \mathscr{O}\left(\frac{J_y}{J_x}\right)^3.
\end{aligned} \tag{16}$$

Our treatment can be readily generalized to the case of an arbitrary $N_y$-leg Hofstadter strip with flux $\Phi$. Indeed, we observe that $\langle \varphi_n | \hat{n} | \varphi_n \rangle = n(1 + \mathscr{O}\left(\lambda^4\right))$ for a state in the bulk, $|n| < \frac{N_y - 1}{2}$, whereas $\langle \varphi_n | \hat{n} | \varphi_n \rangle = n(1 - \left(\frac{\lambda}{\mathscr{E}_{2,1}}\right)^2 + \mathscr{O}\left(\lambda^4\right))$, with $\mathscr{E}_{2,1} = 2(1 - \cos\Phi)$, for a state in the edge, $|n| = \frac{N_y - 1}{2}$.

Thus, at the crossings of the dispersion relation, the band gaps open at first order in $\lambda$, such that $\Delta E_j \propto J_y$. The hybridization gaps, however, only open at order $\lambda^2$. We conclude that the

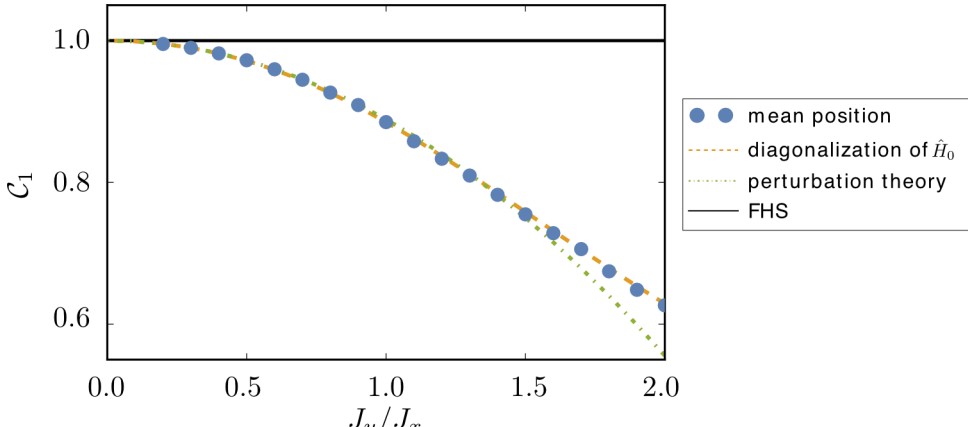

Figure 6:    Measured Chern number as a function of $J_y/J_x$, for $\Phi = \frac{2\pi}{3}$ and $N_y = 3$. We compare the results of mean position measurements, calculated by using Eq. (11), to the ones obtained by diagonalizing $\hat{H}_0(k_x)$, Eq. (3), by using perturbation theory Eq. (16), and by the FHS algorithm.

hybridization gaps are subleading relative to the gap. This allows us not only to identify edge states –states living in the gap and polarized towards the edge of the synthetic dimension [38]– but also to define bulk topological properties which can be revealed by adiabatic pumping. Assuming that none of the gaps close up when $\lambda$ is varied [7], the result obtained is the same as for a large system pierced by the same magnetic flux $\Phi$.

### 4.3.2    Comparison to the measured Chern number

We now compute the Chern numbers from the mean transverse displacement for a broad range of $J_y/J_x$ values, and plot these as blue dots for $N_y = 3$ in Fig. 6. For these simulations, we used a Hofstadter strip subject to a magnetic flux $\Phi = 2\pi/3$. At the beginning of the loading sequence, the atoms are spin polarized with $y = 0$, and are spatially constrained to a region of $w_x = 30$ sites, then we load the lattice by linearly ramping up $J_y$. At $t = 0$, we apply a force with small amplitude $F_x = 0.02\Delta E_1/d$, such that the single band approximation is applicable.

In Sec. 4.3.1 we calculated analytically the mean spin value of the lowest energy eigenstate of $\hat{H}_0(k_x = \frac{2\pi}{3d})$ using second order perturbation theory. To confirm its validity we diagonalize the Hamiltonian Eq. (3), and plot in Fig. 6 the mean spin value of its lowest energy eigenstate at $k_x = \frac{2\pi}{3d}$ together with the result from perturbation theory. For sufficiently small $J_y/J_x$, both approaches yield the same result, and are also in excellent agreement with our Chern number measurement, as displayed in Fig. 6. We conclude that the coupling between different spin states is responsible for the reduced $y$ displacement which we observe in Fig. 5. Importantly, we see that our data smoothly converge to the expected Chern number $\mathscr{C}_1 = 1$ when $J_y/J_x \to 0$.

It is important to understand that the corrections to the wave packet's displacement which we are observing are in fact an edge effect. Indeed, our measurement of the Chern number relies on a correlation between the atomic quasi-momentum $k_x$ and its mean spin value. This correlation is not destroyed when the bulk eigenstates of $\hat{H}_0$ are delocalized in the synthetic dimension. When the eigenstates have amplitude at the edges of the system, however, delocalization in the spin direction is highly anisotropic. The consequence, as we observed in Sec. 4.3.1, is that their mean position becomes dependent on the tunneling amplitude in the spin direction.

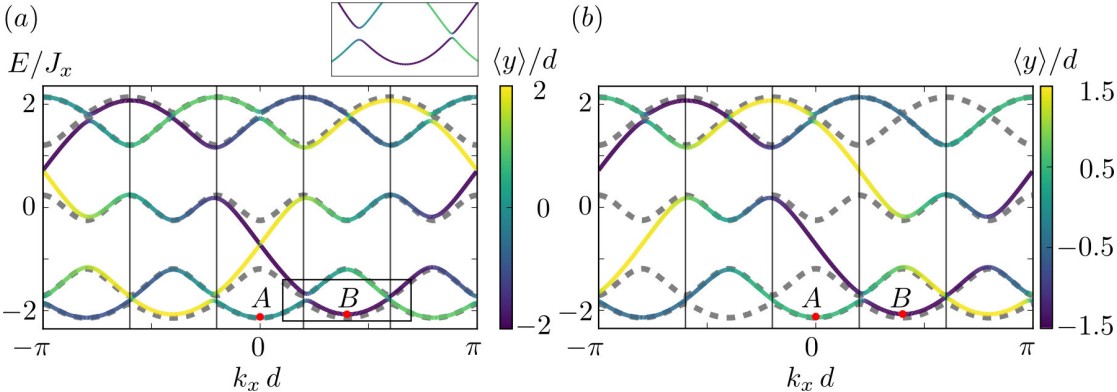

Figure 7: Spectrum of a Hofstadter strip with $\Phi = 4\pi/5$. The coloring of the lines indicate the mean eigenstates' position along $y$ for a system with $N_y = 5$ in panel (a), and with $N_y = 4$ in panel (b). The dashed gray lines indicate in both cases the dispersion relation of a strip with $N_y = 5$ and periodic boundary conditions. The inset in panel (a) highlights a band gap appearing at $k_x = \frac{\pi}{5d}$, and a pair of well localized edge states crossing at $k_x = \frac{3\pi}{5d}$. In the dynamics described in the text, the momentum of the wave packet evolves from point $A$ to $B$.

## 4.4 Measurement of higher Chern numbers

Strikingly, our method can be extended to study Hofstadter strips which present a Chern number $|\mathscr{C}_1| > 1$. In these systems, the Chern numbers calculated using the FHS method can become dependent on the system's size. Despite this, we show that we can still relate our measurement to the Chern number of the infinite system. To this aim, we will choose a flux of $\Phi = 4\pi/5$, which, in the limit of an extended system, yields $\mathscr{C}_1 = -2$. For definiteness, in the remainder of this subsection we will consider a system with $J_y = J_x/2$, which yields a lowest energy gap $\Delta E_1 = 0.11 J_x$. For the adiabatic approximation to be valid, we apply an extremely weak force, with amplitude $F_x = 0.01 \Delta E_1/d$. While this value is comparable to the first hybridization gap, this will not cause any measurement errors because the scheme we suggest in this section does not populate the edge states.

### 4.4.1 Five-leg strip

Let us start by considering a strip with $N_y = 5$, whose dispersion relation is presented in Fig. 7a. The periodic system, plotted in gray, presents five bands. Two pairs of edge states are clearly visible between the first and second bands, which cross at $k_x = \pm\frac{3\pi}{5d}$ (see inset). These can be found either by inspecting the dispersion relation or through analytical calculations [97].

In order to study the pumping dynamics, we initiate our system with a state which occupies $w_x = 50$ sites in the spatial direction, and occupies only the $n = 0$ site in the spin dimension. This initial state corresponds to a superposition of eigenstates narrowly centered around the point $A$ in Fig. 7a. The atomic wave packet is subsequently pumped to the $n = -2$ site of the bottom band during the first period of evolution, which is annotated by point $B$. Importantly, the dynamics takes place without the wave packet leaving the periodic system's bottom band, thereby allowing us to measure this band's Chern number. We do this as described in Sec. 3.1, by measuring the displacement of the atomic center of mass in the $y$ direction, which yields $\mathscr{C}_1 \approx -1.98$, which is in excellent agreement with the expected value.

Let us note here that the detection scheme proposed above would not work if the wave packet was initially prepared at $k_x = -4\pi/5d$, since the following dynamics would displace the momentum density along the dispersion relation through the tiny gap $\propto (J_y/J_x)^2$ located at $k_x = -3\pi/5d$, and the wave packet would be transferred to the next band. The scheme

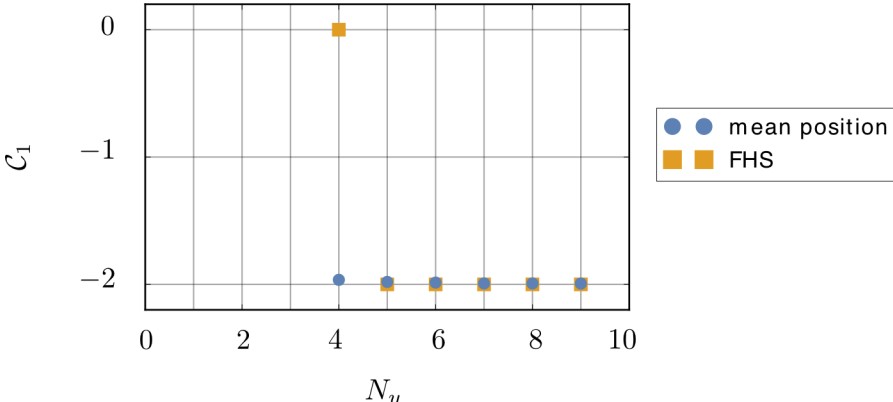

Figure 8: We estimate the lowest band Chern number for $\Phi = 4\pi/5$ and $N_y \in [3, 9]$, measured using atomic pumping (blue dots) and with the FHS algorithm (orange squares). For $N_y > 5$, both results agree, but differ for for $N_y < 5$. This is due to the breakdown of the FHS method.

proposed above would instead work if the wave packet was initialized at $k_x = -2\pi/5d$, since the first gap crossed during the dynamics is much larger, $\propto (J_y/J_x)$, so that the wave packet remains in the ground band during a complete period $T$.

### 4.4.2 Generalization to $q > N_y$: four-leg strips

The Hofstadter strip with $N_y = 4$ (so that $n$ takes one of the four discrete values {-3/2, -1/2, 1/2, 3/2}) and flux $\Phi = 4\pi/5$ is particularly interesting because, at every given quasi-momentum $k_x$, the Hamiltonian $\hat{H}_0(k_x)$, Eq. (3), has a number of eigenstates which is smaller than the number of bands of the corresponding extended model (i.e., $N_y < q$). The dispersion relation of this system is plotted in Fig. 7b.

Interestingly, it is still possible to measure the lowest band's Chern number, provided we ensure that all of the dynamics takes place in this band. As previously, we load the lattice in the bottom band with an atomic wave packet which occupies $w_x = 50$ sites in the $x$ direction, and is completely localized at $n = 1/2$. It is therefore in a superposition of states narrowly centered around the point $A$ in Fig. 7b.

By applying the constant force to the system, we pump the atomic density adiabatically from the $n = 1/2$ site (point $A$) to the $n = -3/2$ site (point $B$). As can be seen by inspecting the dispersion relation, all the eigenstates explored during this pumping sequence can be associated to the bottom band of the periodic system. This means that, in this setting, we are again able to measure the first band's Chern number from the atomic displacement, yielding the value of $\mathcal{C}_1 \approx -1.96$.

### 4.4.3 Comparison to the FHS algorithm results

Using the same initial state as in the previous subsections, we measure the Chern number in this system for $N_y \in [4, 9]$. These are plotted as blue dots in the Fig. 8. As $N_y$ is increased, we observe that the measured value tends asymptotically to $\mathcal{C}_1 = -2$.

Alongside these estimates, we plot with orange squares the Chern number calculated using the FHS algorithm. When $N_y < 5$, the latter deviates from the Chern number of the infinite systems. This is simply because, for very small systems, the Berry flux (the integral of the Berry curvature) through some plaquettes can exceed $\pm\pi$, such that the FHS algorithm cannot record them accurately [95]. It is interesting to observe, however, that the Chern number measured

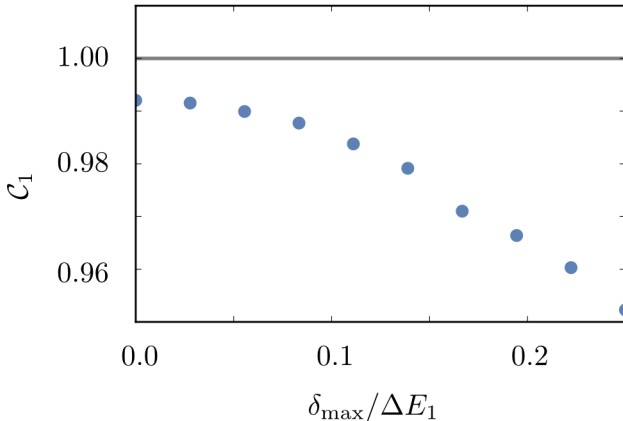

Figure 9: Chern number of the ground band measured for $\Phi = 2\pi/3$ and $N_y = 3$, in presence of static onsite disorder in both the spatial and synthetic dimensions, with amplitude $\delta_{\max}$. Each data point is an average of 20 realizations.

through the wave packet's transverse displacement still yields a very good estimate of the extended lattice value even when $N_y = 4$. We conclude that the Chern number calculated using the FHS algorithm is not related to the system's transverse conductance in this limit.

## 5 Robustness of the method

In this Section, we study the robustness of our measurement against two types of experimentally relevant perturbations: static spatial disorder and harmonic trapping.

### 5.1 Static disorder

A fundamental property of topologically non-trivial systems is their robustness to local perturbations. The study of the interplay of disorder and topological phases is an extremely rich area of research [98, 99].

To understand to which extent disorder may affect our scheme, let us consider a Hofstadter strip in presence of random onsite energy shifts

$$\hat{H}_{\text{dis}} = \hat{H} + \hat{V}_{\text{dis}} = \hat{H} + \sum_{m,n} \delta_{m,n} \hat{c}^{\dagger}_{m,n} \hat{c}_{m,n}, \tag{17}$$

where $\delta_{m,n}$ is uniformly distributed in the interval $[0, \delta_{\max}]$ and corresponds to uncorrelated disorder in both the spatial and synthetic dimensions. We take $N_y = 3$, a magnetic flux of $\Phi = 2\pi/3$, and $J_y/J_x = 1/5$. The system is then adiabatically loaded with spin polarized atoms in the $n = 0$ state, constrained to $w_x = 60$ sites in the $x$ direction. The band gap has amplitude $\Delta E_1 = 1.65 J_x$, and we subject the atoms to the constant force $F_x = 0.03 \Delta E_1/d$.

In Fig. 9, we plot the measured Chern number of the lowest band, averaged over 20 realizations, as a function of $\delta_{\max}$. For increasing amplitudes of the random potential, the measurement of the Chern number deviates steadily from the expected result, and we identify the broadening of the wave packet as the main source of error. Remarkably, even for disorder amplitudes as high as $\delta_{\max} = \Delta E_1/4$, we still measure a Chern number which is within 94% of its actual value. We conclude that, as is generally the case in topological systems, disorder does not significantly affect topological measurements in Hofstadter strips up to disorder amplitudes comparable to the band gap.

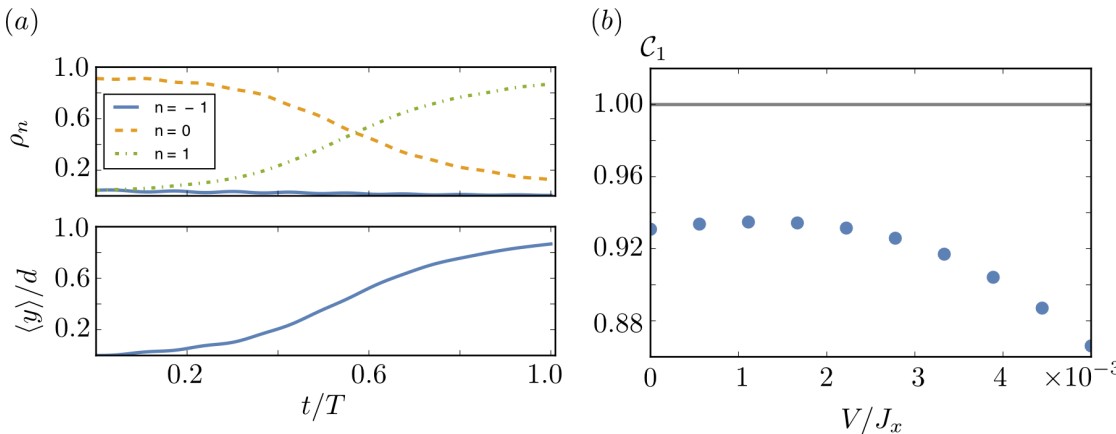

Figure 10: Pumping dynamics in a trap. We consider the case of a flux $\Phi = \frac{2\pi}{3}$ and $N_y = 3$. (a) Spin populations $\rho_n$ (top panel) and mean displacement along $y$ (bottom panel), plotted as a function of time, in a trap of strength $V = 5 \times 10^{-3} J_x$. (b) Measured ground band Chern number as a function of the trap depth $V$.

## 5.2 Harmonic trap

In ultracold atom experiments the gas is contained in a harmonic trap. The Hamiltonian for atoms of mass $M$ then reads

$$\hat{H}_{\mathrm{ho}} = \hat{H} + \hat{V} = \hat{H} + V \sum_{m,n} (m - N_x/2)^2 \hat{c}^\dagger_{m,n} \hat{c}_{m,n}, \tag{18}$$

where $V = M\omega^2 d^2 / 2$ denotes the trap characteristic energy. To ensure that this potential does not disturb our measurement, we must ensure that the corrections to the dynamics it induces are negligible on the time scale of a Bloch oscillation, $T$. Specifically, the force might become spatially dependent, blurring some observables [100], or even inducing dipole oscillations competing with the Bloch oscillations [101] on which our protocol relies. To limit these detrimental effects, we impose the experimental condition $\omega/2\pi \ll 1/T$, therefore placing a lower bound on the force $F_x$.

For concreteness, let us consider $^{41}$K atoms in an optical lattice with lattice spacing $d = 532$ nm. Typical spatial tunneling amplitudes and trap frequencies correspond to $J_x/h \sim 100$ Hz and $\omega/2\pi \sim 30$ Hz, leading to characteristic trapping energies on the order of $10^{-3} J_x$.

Let us consider $V = 5 \times 10^{-3} J_x$ and estimate the Chern number for a wave packet in a lattice with $N_y = 3$, subject to an external magnetic flux of $\Phi = 2\pi/3$, $J_y/J_x = 0.7$ (corresponding to a band gap $\Delta E_1 = 1.65 J_x$), and a force $F_x = 0.1 \Delta E_1/d$, giving $T^{-1} \approx 50$ Hz $\gg \omega/2\pi$.

As in previous sections, we adiabatically load the lattice with a wavefunction spin polarized with $n = 0$. Due to the presence of the harmonic trap, however, we do not need to spatially constrain the initial state along $x$ to obtain a well localized state. In Fig. 10a, we plot typical spin populations $\rho_n, n \in \{-1, 0, 1\}$ and the mean displacement along $y$ as a function of time. From the mean $y$ displacement over a period we obtain an estimate of the Chern number $\mathscr{C}_1 = 0.87$. Once again this is in very good agreement with the expected value of 1.

As shown in Fig. 10b, the extracted Chern number only weakly depends on the characteristic trap energy $V$, proving that the proposed measurement of the Chern number is robust to a wide range of trap amplitudes. This study shows that our protocol is experimentally realistic, and can provide an accurate measurement of the Chern number with present day technology.

# 6   Conclusions and Outlook

In this work, we showed that, somewhat counterintuitively, quantized topological features are present even when an extended 2D lattice is drastically reduced in size in one direction. In particular, we proposed a scheme to explore the topological properties of Hofstadter strips. The method relies on three main ingredients: the adiabatic loading of an atomic wave packet well localized in the short direction in the ground state of the lattice, the application of a force in the long direction, and the measurement of the center-of-mass position after a Bloch oscillation.

Despite the limited extension of the strip in one direction, we showed that the transverse displacement of the center of mass of a wave packet prepared in any given band converges, in the limit of small transverse tunneling, to the corresponding quantized Chern number of the extended system. We discussed how the hybridization of edge states affects the measurement of the Chern number and showed that the latter can be quantitatively characterized with second order perturbation theory. We showed that our detection scheme can also probe Chern numbers of the ground band which are larger than one. Remarkably, our protocol remains valid even for strips that are so narrow along one direction that the FHS algorithm breaks down. In order to test the experimental feasibility of the protocol, we verified that an accurate measurement of the Chern number is also possible in presence of disorder, and for realistic harmonic traps.

Hofstadter strips are presently of great experimental relevance, given that ultracold atoms in synthetic lattices naturally realize such geometries. In this case, the transverse displacement corresponds to the mean spin polarization, which can be directly read out in standard time-of-flight measurements. Nonetheless, the physics discussed here may be probed in other, very different setups, such as in photonic or optomechanical platforms. This work can be readily generalized to measure Chern numbers in other strips geometries [52, 68] and $Z_2$ invariants of $\mathcal{T}$-symmetric strip insulators [3, 86]. Alternatively, the effect of strong interactions on our protocol could also be analyzed [61, 70], building on recent studies of interacting phases and Laughlin-like states in real and synthetic ultracold atom ladders [102–105]. Our study suggests a relation between large systems and their strip geometry counterparts. It would be interesting to find out if this relation extends to other classes of topological insulators in two and higher dimensions. Our method could, for instance, be extended to measure the second Chern number in synthetic quantum Hall systems with a short extra dimension. This invariant has been measured recently in optical lattices by exploiting a 2D extension of the Thouless pump [106].

## Acknowledgements

We acknowledge Dina Genkina and Ian Spielman for enlightening discussions.

**Funding information:**   This work has been supported by Spanish MINECO (SEVERO OCHOA Grant SEV-2015-0522, FISICATEAMO FIS2016-79508-P and StrongQSIM FIS2014-59546-P) the Generalitat de Catalunya (SGR 874 and CERCA program), Fundació Privada Cellex, DFG (FOR 2414), and EU grants EQuaM (FP7/2007-2013 Grant No. 323714), OSYRIS (ERC-2013-AdG Grant No. 339106), QUIC (H2020-FETPROACT-2014 No. 641122), SIQS (FP7-ICT-2011-9 Grant No. 600645) and MagQUPT (PCIG13-GA-2013 No. 631633). This project was also supported by the University of Southampton as host of the Vice-Chancellor Fellowship scheme. A.D. is financed by a Cellex-ICFO-MPQ fellowship. P.M. and L.T. acknowledge funding from the "Ramón y Cajal" program.

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
