# Peer review of "Measuring Chern numbers in Hofstadter strips"

_SciPost Physics, doi:SciPost Phys. 3, 012 (2017)_

## Round 1 · Referee Report · Anonymous (Referee 1) · 2017-6-11

Strengths

  1. The authors propose and analyse a novel method enabling to determine topological properties of the bulk by measuring the dynamics of an wave-packet in a Hofstadter stripe.
  2. This is can be useful, e.g., to cold atom experiments where the magnetic flux can be produced using synthetic dimensions. In that case the stripe naturally appears, because the synthetic dimension extends over a limited number of sites.

Weaknesses

  1. The manuscript deals only with the single particle physics. I suppose the many-body aspects of this method will be studied later on, as in the Concluding Section VI it is written: "Another interesting direction is to analyze the effect of strong interactions on our protocol [60, 69], specially in view of the recent theoretical interest on interacting phases and Laughlin-like states in real and synthetic ultracold atom ladders [97–100]."

Report

The authors consider the transport of particles in two-dimensional lattices affected by a uniform magnetic. It is shown that the topological properties of the bulk can be determined from the dynamics of an wave-packet in a Hofstadter stripe taken from the bulk. This is relevant for example to cold atom experiments where the magnetic flux can be produced using synthetic dimensions. In that case the stripe naturally appears, because the synthetic dimension extends over a limited number of sites.

The authors have explored the dynamics of a wave-packet which is initially localized in the transverse direction of the stripe. In the context of the semi-synthetic lattices such a wave-packet can be produced by initially placing atoms in a single internal state: this corresponds to a perfect localization in the synthetic dimension. Subsequently a weak constant force is applied in the real dimension leading to the Bloch oscillations. It is shown that after one Bloch cycle, the wave-packet acquires a displacement in the transverse (synthetic) direction determined by the quantized Chern number of the infinite system. The scheme enables to determine the Chern numbers of both the ground and excited bands. The robustness of the proposed method with respect to the disorder and harmonic trapping has also been investigated.

The paper is well written. I think it is an interesting work, which is relevant to the current experiments with cold atom, photonic and condensed matter systems. For example, applying the proposed method the Chern number of the bulk can be measured using the stripes containing as few as three sites in transverse direction. This could be very helpful in studying the semi-synthetic ribbons. I recommend the manuscript to be published.

A minor remark. The method considered by the authors enables one to measure the first Chern number. The authors could discuss the extension of the method to measure of the second Chern number. In the last sentence of the Concluding Section VI there is some hint in this direction, but the authors could be more explicit.

Requested changes

See the end of the Report

  • validity: high
  • significance: high
  • originality: high
  • clarity: high
  • formatting: good
  • grammar: good

Author:  Alessio Celi  on 2017-06-15  [id 144]

(in reply to Report 1 on 2017-06-11)
Category:
remark

We thank the Referee for the interest in our work.
The suggestion of the Referee of discussing the extension of our method to the measurement of the 2nd Chern is very interesting and timing. We will certainly comment more on this point in the conclusions in the revision of the manuscript. We will do it also in relation to the very recent experiment in Bloch group where the 2nd Chern number is measured through a 2D Thouless pump, experiment appeared little after the submission of the present manuscript.

---

## Round 1 · Referee Report · Anonymous (Referee 2) · 2017-6-21

Strengths

  1. The authors propose a novel way to extract the topological Chern number from the transport of a wave-packet in a narrow Hofstadter strip. This is a very timely and interesting proposal, which opens up new perspectives for measuring topological invariants in small systems.

  2. This proposal seems realistic from an experimental point-of-view, and it nicely exploits the advantages of using synthetic dimensions in ultracold atoms. The authors also support their proposal by discussing the robustness of their method under harmonic trapping and disorder.

  3. This paper is clearly explained and should be accessible to many people in the field. It also presents an understanding of Chern number physics which is quite intuitive while also being complementary to the current literature.

  4. The strength of this method is further demonstrated by its application to measure higher Chern numbers and to even beat the FHS algorithm, which is a much used tool in the field.

Weaknesses

  1. There are a few issues that need to be clarified within the manuscript. These are listed as Requested Changes below.

Report

In this paper, the authors propose to measure topological Chern numbers from the dynamics of a wave-packet in a narrow Hofstadter strip: a two-dimensional lattice, pierced by a uniform magnetic flux, which is restricted to only a few sites along one of the spatial directions. This proposal is directly relevant to cutting-edge ultracold gas experiments on so-called synthetic dimensions built out of internal atomic states. Due to practical limitations on the number of internal atomic states coupled, these experiments realised Hofstadter strips and explored quantum Hall edge physics; this paper now proposes how to go further to also probe the topological bulk properties associated with a fully-extended 2D system. This is therefore a timely and interesting proposal, which can be helpful to experimentalists in the field.

Overall, this paper is well-written, accessible and well-explained. The authors have also cited widely, so that I think this paper can also serve as a valuable starting point for an interested reader to explore the rest of the field. I recommend publication of this paper after a few small issues, listed in Requested Changes, have been resolved.

Requested changes

  1. When introducing synthetic lattices on pg. 2, the authors specify "systems where particles have D spatial degrees of freedom and an extra synthetic dimension" but then later in the paragraph talk about "4D models", which implies another definition of "D"?

  2. Refs. [45-47] are cited as experiments on pg. 2, but these are theoretical proposals. I would recommend adding these instead to "...optical resonators [49]".

  3. In Figure 1a) the blue arrow is very hard to see. I would also suggest adding axes labels for Fig 1.b) to be clear. It would also be helpful to the reader to explain the coloring of different sites in Fig. 1b).

  4. As a question to the authors: how can we see that all quasi-momentum states are populated equally over this definition of a period of the force (Eq. 10)? From Section II B and as implied by Fig. 3, the chosen Brillouin zone is $(2 \pi)/d \times (2 \pi/q d)$ along $k_x$ and $k_y$ respectively, which implies that you need to wait $T=2 \pi \hbar / d |F_x|$ to change $k_x$ by $2\pi$ and to sweep through the whole MBZ?

  5. Could the authors please comment e.g. in Sec IV A on how they choose the force relative to the hybridisation gap? What size is the hybridisation gap here?

  6. In Sec IV A, the authors say "the Hamiltonian with open boundary conditions and zero force, Eq. (3)". Do they mean Eq. 1?

  7. Could the authors comment on how, when they project the evolved state on the eigenvectors of the Hamiltonian with open boundary conditions, they make the distinction between band and gap states in Fig 4 b)?

  8. In Sec. IV C1, the authors refer to "point A in Fig. 2a" but I do not see point A in this figure?

  9. What is the strength of the force used in Section IV D? In the inset of Fig. 7, the hybridisation gap looks very close to the energy band gap so it is possible to chose a good value for the force between these limits?

  10. In Fig. 7, I am curious about the identification of "well localised edge states crossing at $k_x = 3 \pi / 5d$": according to the color scheme, I see that the state with a positive gradient is well-localised at $<y>=-2d$, but the state with a negative gradient does not seem to be localised as I would have expected around $<y>=2d$. Why can this second state be called an edge state?

  11. Question to the authors: in Fig. 7 & 8, it is shown that the Chern number of the lowest band can be extracted even when the number of eigenstates is smaller than the number of bands in the corresponding extended model. If instead the experiment was started from the minimum at $k_x=-4 \pi/5d$ then would this still work? Could the authors comment on how general these results are expected to be?

---

## Round 2 · Author Response

Dear Editor,
Thanks for sending us the referee report of our manuscript.
We would like to thanks the referees for the interest in our work and the careful reading of the manuscript. We have addressed all the points raised by the referees, see below for details.
We have also changed the manuscript style to the SciPost one by following the Author guideline.
With best regards,
The Authors

---

## Round 2 · List of Changes

We indicate the requested changes by the Referees with numbers and our answer/change description with #.
* * *
* * *
Reply to Referee A
* * *
* * *
We thank the Referee for his/her careful reading of the manuscript, positive reply, and valuable comments and suggestions. Below, we discuss the requested change.

1. A minor remark. The method considered by the authors enables one to measure the first Chern number. The authors could discuss the extension of the method to measure of the second Chern number. In the last sentence of the Concluding Section VI there is some hint in this direction, but the authors could be more explicit.
#. The suggestion of the Referee of discussing the extension of our method to the measurement of the 2nd Chern is very interesting and timing.
We have added the following comment to the conclusions:
"Our method could, for instance, be extended to measure the second Chern number in synthetic quantum Hall systems with a short extra dimension.
This invariant has been measured recently in optical lattices by exploiting a 2D extension of the Thouless pump [105].",
where the reference 105 refers to the very recent experiment in Bloch group appeared little after the submission of the present manuscript.
* * *
Reply to Referee B
* * *
We thank the Referee for his/her careful reading of the manuscript, positive reply, and valuable comments and suggestions.
We believe that by answering his/her questions we have improved the quality of our manuscript.
Below, we discuss the requested changes one by one, and we explain the corresponding changes we have performed to the manuscript.

1. When introducing synthetic lattices on pg. 2, the authors specify "systems where particles have D spatial degrees of freedom
and an extra synthetic dimension" but then later in the paragraph talk about "4D models", which implies another definition of "D"?
#. We agree with the referee's comment. To remove the ambiguity, we have now rephrased the text as "... lattices which have a synthetic dimension,
which is obtained by coherently and sequentially coupling particles' internal degrees of freedom."

2. Refs. [45-47] are cited as experiments on pg. 2, but these are theoretical proposals. I would recommend adding these instead to "...optical resonators [49]".
#. We agree, and we have followed the referee's suggestion.
2bis. In Figure 1a) the blue arrow is very hard to see. I would also suggest adding axes labels for Fig 1.b) to be clear. It would also be helpful to the reader to explain the coloring of different sites in Fig. 1b).
#. We agree, and we have followed the referee's suggestions.
Regarding the coloring of the sites in Fig.1(b), we'd rather not refer to spin states here, because actually what we discuss would be true also for real lattices.

3. As a question to the authors: how can we see that all quasi-momentum states are populated equally over this definition of a period of the force (Eq. 10)? From Section II B and as implied by Fig. 3, the chosen Brillouin zone is $(2?)/d \times (2 ?/q d)$ along kx and ky respectively,
which implies that you need to wait $T=2? \hbar/d|Fx|$ to change kx by 2? and to sweep through the whole MBZ?
#. We thank the referee for raising these points and stimulating us to comment on them.
We address first the equal population after one period the force, and then the influence of the gauge fixing on the measure of the Chern number.
For the first point, we assume that the initial state occupies uniformly ky and that the motion follows the semi-classical equations,
in particular Eq. (5).
Therefore, after one period of the force the wave packet has swept the whole Brillouin zone.
To clarify this last point, we added the following phrase just after Eq. (9):
"Since the probability density displaces with uniform velocity, the mean Bloch state occupation over a period of the force is simply the uniform distribution,"
For the second point, let us first comment that physical quantities such as the energy spectrum
and the Chern number are gauge invariant and therefore do not depend on the gauge fixing.
In particular, as nicely discussed in Appendix B of Ref. [89] (of the revised version of the manuscript),
this implies that the energy spectrum is completely defined on the so called "reduced magnetic Brillouin zone",
defined as -\pi/(qd) \le kx , ky <\pi/(qd).
A direct consequence of the latter is that the normal group velocity of the semi-classical equations of motion will have a period of 2\pi/(qd).
Let us now discuss the case of the anomalous velocity in the semi-classical equations of motion. A priori, the Berry curvature is a gauge invariant quantity. However, as discussed in Refs [90,91] (of the revised version of the manuscript), unlike the energy spectrum, the value of the Berry curvature depends on the definition of the Fourier transform.
This can lead to unphysical effects where the Berry curvature seems to depend on the gauge and loses the symmetry of the lattice.
This would be in contraction with any physical observable which should depend neither on the Fourier transform, nor on the gauge choice.
However, as discussed in these works, there always exists one definition of the Fourier transform that recovers the symmetry of the Berry curvature
with respect to the lattice symmetry and therefore corresponds to the semiclassical equations of motion.
In the case of the Hofstadter model, the Berry curvature is also completely defined on the reduced magnetic Brillouin zone.
Therefore, after a period of 2\pi/(qd), the system has performed a whole Bloch oscillation,
which results in a displacement in the direction perpendicular to the force of a number of sites equal to the Chern number $C$.
Now, coming back to the different gauges, we can see that there are no contradictions in the case of periodic boundary conditions.
In the Landau gauge with the gauge field along y, after a period of the force, i.e. 2\pi/(qd),
the atomic center of mass has jumped $C$ sites in the y direction, which really corresponds to having the unit cell in real space coming back to itself.
In the Landau gauge with the gauge field along x, for the unit cell to come back to itself,
one has to sweep the momentum distribution over 2 \pi/d, which corresponds to a jump of $q*C$ sites.
However, as discussed before, in this gauge this will correspond to q jumps of $C$ sites at displacements of the momentum distributions of 2\pi/(qd).
To clarify this point, we have specified explicitly the gauge used in the derivation in Sec. IIIA,
we have included Refs. [89,90,91] in the bibliography, and we have added a paragraph at the end of Sec. IIIA:
"We have derived the mean displacement after one pumping period, given by Eq. (11), in the gauge of Eq. (1),
the Landau gauge with the gauge field along the y direction. Naively, we could expect this relation to change if we set the gauge field along x instead.
As discussed in Refs. [89-91], due to the symmetry of the lattice, both the energy spectrum and the Berry curvature are completely defined in the so-called reduced magnetic Brillouin zone -\pi/(qd)\le kx , ky < \pi/(qd). Hence, during one complete period of the force, $T \propto 2\pi \hbar/(qd|Fx|)$, the wavepacket explores the reduced magnetic Brillouin zone in its entirety, thereby performing a complete Bloch oscillation. This is true regardless of the gauge choice. Thus, we see that the relation between the mean displacement and the Chern number, as given by Eq. (11), is independent of the gauge choice, as is always the case for physical quantities."

4. Could the authors please comment e.g. in Sec IV A on how they choose the force relative to the hybridisation gap? What size is the hybridisation gap here?
#. We chose the force to be large with respect to the hybridisation gap to allow the pumping to higher energy bands and small with respect to the energy gap to avoid Landau-Zener transitions.
This is indeed possible since the energy gap scales as \lambda=Jy/Jx and the hybridization gap scales as \lambda^2, as discussed in Sec. IVc1.

5. In Sec IV A, the authors say "the Hamiltonian with open boundary conditions and zero force, Eq. (3)". Do they mean Eq. (1)?
#. We have clarified this point, writing:
"[The populations] were obtained by projecting the evolved state on the eigenvectors of the Hamiltonian, Eq. (3)."

6. Could the authors comment on how, when they project the evolved state on the eigenvectors of the Hamiltonian with open boundary conditions, they make the distinction between band and gap states in Fig 4 b)?
#. To clarify this point, we have modified the text, adding the sentence
"We define the lower (upper) gap as the separation between bands at kx=\pi/(3d) (kx=2\pi/(3d)), and assume that whatever lies outside the gaps belongs to the corresponding band."

7. In Sec. IV C1, the authors refer to "point A in Fig. 2a" but I do not see point A in this figure?
#. We agree. The sentence refers now to "point A in Fig. 4a."

8. What is the strength of the force used in Section IV D? In the inset of Fig. 7,
the hybridisation gap looks very close to the energy band gap so it is possible to chose a good value for the force between these limits?
#. In this section, we wanted to prove that our method allows the measurement of Chern numbers of the lowest energy band greater than $1$.
We therefore chose a force small with respect to the energy gap but without any constraint with respect to the hybridization.
Indeed, we only study the trajectory between points A and B, that are bulk states and do not involve topological edge states.

9. In Fig. 7, I am curious about the identification of "well localised edge states crossing at kx=3?/5d: according to the color scheme,
I see that the state with a positive gradient is well-localised at <y> = -2d, but the state with a negative gradient does not seem to be localised as I would have expected around <y> = 2d. Why can this second state be called an edge state?
#. When $C>1$, the edge states are not always located on the first and last site of the cylinder.
As nicely discussed in Ref. [96] (that we have added to the bibliography), one can have edge states at 1,L and 2, L-1.
In the commensurate case (that can be computed analytically), i.e. L=q*l-1, the edge states will come in pairs located at sites 1,L and edge states located at sites 2,L-1.
However, in the incommensurate case, the edge states can come in any type of pairs.
Therefore, an edge state is defined as a state located in the energy gap of the bulk and which is current-carrying.
Upon increasing the size of the system, edge states become exponentially located at the edge of the system.

10. Question to the authors: in Fig. 7 & 8, it is shown that the Chern number of the lowest band can be extracted even
when the number of eigenstates is smaller than the number of bands in the corresponding extended model.
If instead the experiment was started from the minimum at kx=-4?/5d then would this still work? Could the authors comment on how general these results are expected to be?
#. The referee touches an important point here. We have added a sentence to the manuscript to clarify it:
"Let us note here that the detection scheme proposed above would not work if the wavepacket was initially prepared at kx=-4\pi/(5d),
since the following dynamics would displace the momentum density along the dispersion through the tiny gap \propto(Jy/Jx)^2 located at kx=-3\pi/(5d),
and the wavepacket would be transferred to the next band. The scheme proposed above would instead work if the wavepacket was initialized at kx=-2\pi/(5d),
since the first gap crossed during the dynamics is much larger, \propto(Jy/Jx), so that the wavepacket remains in the ground band during a complete period $T$."

You are currently on this page

Resubmission 1705.04676v2 on 21 July 2017

---

## Editorial Decision

published